# Validation of the Polar Fitness Test for Estimation of Maximal Oxygen Consumption at Rest in Medically Supervised Exercise Training: Comparison with CPET and the 6-Minute Walk Test [note 1]

**DOI:** 10.3390/s25185649

**Published:** 2025-09-10

**Authors:** Michael Neudorfer, Lukas Ötzlinger, Devender Kumar, Josef Niebauer, Jan David Smeddinck, Mahdi Sareban, Gunnar Treff

**Affiliations:** 1University Institute of Sports Medicine, Prevention and Rehabilitation, University Hospital Salzburg, Paracelsus Medical University, 5020 Salzburg, Austria; 2Ludwig Boltzmann Institute for Digital Health and Prevention, 5020 Salzburg, Austria; 3Institute of Molecular Sports and Rehabilitation Medicine, Paracelsus Medical University, 5020 Salzburg, Austria

**Keywords:** telemedicine, mHealth, wearables, photoplethysmography, cardiac rehabilitation, prevention

## Abstract

The Polar Fitness Test (PFT) estimates maximal oxygen consumption ($\dot{V}$O_2_max) under resting conditions using heart rate data from the manufacturer’s wearable devices. We aimed to validate the PFT in a population with cardiovascular risk factors and to compare its results with five established equations predicting $\dot{V}$O_2_max based on the 6-min walk test (6MWT). Twenty-four participants (9 female; age 57.4 ± 10.2 years) undergoing medically supervised exercise training—including seven individuals on heart rate-limiting medication—completed the PFT, 6MWT, and cardiopulmonary exercise testing (CPET), which served as the criterion $\dot{V}$O_2_max measurement. The PFT showed a mean absolute percent-age error (MAPE) of 13.7%, an intraclass correlation coefficient (ICC) of 0.743, a mean bias of −1.0 mL/min/kg, and limits of agreement (LoA) of ±11.4 mL/min/kg compared to CPET. Among the 6MWT-based equations, only the Porcari equation demonstrated similar performance (MAPE 12.6%, ICC 0.725, mean bias 0.2 mL/min/kg, LoA ± 9.7 mL/min/kg), while other equations showed larger errors and systematic deviations. Our data indicate that the PFT may present an easily accessible option to estimate $\dot{V}$O_2_max on population level when exercise-based testing is not feasible. However, its variability limits use for individual clinical decisions, reaffirming the relevance of CPET for accurate assessment.

## 1. Introduction

The maximal volume of oxygen consumed per minute ($\dot{V}$O_2_max) is an important marker of cardiovascular function and cardiorespiratory fitness (CRF) [1]. It is applied over the whole range of human fitness from elite endurance sports to patients with various medical conditions [1], where it is the accepted criterion measure of CRF [2]. CRF is a strong and independent predictor of cardiovascular diseases (CVDs) and mortality [3] and is inversely associated with all-cause mortality [4,5,6]. Notably, CRF is modifiable by training interventions [7] and increased CRF is associated with numerous cardiovascular and non-cardiovascular benefits [4,8].

Furthermore, scientists and medical societies recommend to base risk stratification and personalization of training interventions on CRF in CVD patients [9,10]. Also, in prehabilitation, which is an emerging concept for optimizing outcomes of surgical or chemotherapeutic interventions, CRF evaluation aids in estimating pre-surgical risk [11] and supports personalized exercise prescriptions [12,13].

The standard methodology for assessing $\dot{V}$O_2_max is a cardiopulmonary exercise test (CPET) on a cycle ergometer or treadmill until maximal voluntary exhaustion, during which ventilation and gas exchange are measured [2,14]. However, measuring $\dot{V}$O_2_max requires trained personnel, laboratory facilities, devices, is time-consuming and incurs additional cost factors that restrict its use to specialized centers [2]. Additionally, achieving maximal voluntary exhaustion on a treadmill or cycle ergometer may not be feasible for some patients due to, e.g., exercise intolerance, neurological, or orthopedic limitations [15,16,17].

Thus, $\dot{V}$O_2_max is frequently estimated indirectly through submaximal exercise protocols such as 6-min walk testing (6MWT) [18] or by wearable technology [12,19]. Wearable devices for tracking physical activity and fitness, including $\dot{V}$O_2_max or other metrics—are rapidly advancing [12]. In 2023, global shipments of wearable devices, mainly fitness trackers and smartwatches, exceeded 500 million units and this number is expected to grow by at least 25% by 2028 [20].

One example for a wearable-based $\dot{V}$O_2_max estimation method is the Polar Fitness Test (PFT) [21]. The PFT is an algorithm implemented across a range of Polar wearable devices. It is among the few available approaches that claim to provide a valid $\dot{V}$O_2_max estimation at rest, without requiring physical exertion, solely based on a heart rate measurement and few personal data, thus offering a low-risk, easily accessible alternative to CPET, suitable for both in-person prehabilitation and rehabilitation settings and telerehabilitation [19]. According to the manufacturer, who did not publish the original validation data, the PFT was designed for healthy adults aged 15 to 65 years [21]. A recent systematic review showed that the published PFT validation studies have been conducted in healthy and young adults aged from 20.3 ± 1.9 to 26.0 ± 3.1 years. The authors of the review calculated an average overestimation of $\dot{V}$O_2_max of 2.2 mL/min/kg in this young and healthy populations, with wide limits of agreement (−13.1 to 17.4 mL/min/kg) and therefore a low accuracy on the individual level [19]. It thus remains unclear whether the PFT demonstrates sufficient validity at the population level, and whether it provides the individual-level accuracy required for medical decision-making in rehabilitation settings [22,23]. Furthermore, to date, no scientific studies have specifically examined its validity in individuals over 60 years of age, particularly those with CVD or those taking heart rate-limiting medications.

For patients with various medical conditions, the 6MWT is a well-established surrogate for assessing CRF in clinical settings and does not require a lab or expensive equipment [24]. The distance covered—along with other metrics, depending on the equation applied—is then used to estimate $\dot{V}$O_2_max based on regression equations derived from previous studies. In contrast to the 6MWT, which requires additional time for preparation, walking, recovery and patient safety precautions, the PFT can be completed within 5 min under resting conditions, even without the patient’s ability and motivation to walk.

The objective of the study was to analyze the accuracy of the PFT and its potential for application in rehabilitation settings. Therefore, we validated the PFT in a considerably older population than in previous validation studies. Our participants attended a medically supervised exercise training program. In addition, five previously published equations for estimating $\dot{V}$O_2_max from 6MWT performance were applied to enable comparison with the PFT and CPET derived data.

## 2. Materials and Methods

### 2.1. Study Design

This prospective validation study was conducted in accordance with the STARD (Standards for Reporting of Diagnostic Accuracy Studies) guidelines [25]. Thirty-two participants performed a PFT with a Polar Vantage M2 photoplethysmography (PPG)-based optical heart rate sensor, a 6MWT and a CPET until maximal exhaustion on a treadmill. The ethical board of the State of Salzburg approved the study protocol, informed consent forms and all supplementary study documents (No. 1017/2022). The study was performed in compliance with the Declaration of Helsinki and its current amendments [26] and was prospectively registered at ClinicalTrials.gov (NCT05525000).

### 2.2. Participants

We invited healthy hospital employees and patients with diverse underlying health conditions, all of whom were either currently attending or intending to attend a medically supervised exercise training program at the University Institute of Sports Medicine, Prevention and Rehabilitation of the University Hospital Salzburg to participate in the study. Inclusion criteria comprised an age of 35 to 80 years, while exclusion criteria encompassed signs of acute infections or diseases, atrial fibrillation, clinical signs of chronic venous insufficiency ≥ grade 1 at upper extremities, physical limitations for carrying out the planned examinations and tests, skin wounds or tattoos at sensor placement site, or pregnancy.

The sample size for this study aligns with that of the main research question, which focused on examining the validity and reliability of optical heart rate sensors [27]. However, we performed post hoc sample size calculation for the current research question, applying the formula proposed by Bujang et al. [28] to estimate the required number of subjects based on the intraclass correlation coefficient (ICC). We defined an expected ICC of 0.90, as reported in previous studies on similar comparisons [19], and set a minimum acceptable ICC of 0.75, corresponding to “good” reliability according to Koo and Li [29]. The significance level (α) was set at 0.05, and statistical power (1–β) at 0.8. Using this approach, the required sample size was calculated to be 23 participants. Our final study included 24 participants, thereby meeting this requirement.

### 2.3. Study Protocol

#### 2.3.1. Medical Examination

Prior to data collection, informed consent was obtained from all participants. All study examinations were performed at the University Institute of Sports Medicine, Prevention and Rehabilitation, University Hospital Salzburg, Salzburg, Austria. A single senior physician screened the participants for inclusion and exclusion criteria. In case of inclusion, skin type was assessed using the Fitzpatrick-Score [30].

#### 2.3.2. Sensor Placement and PFT

Before starting the test protocol, the Polar Vantage M2 sensor was reset for each participant. A new user account was created for each individual using the Polar Flow© desktop application, which served as the interface for storing all data recorded by the Polar Vantage M2. This Polar sensor uses the Polar Precision Prime™ technology, which combines multiple LEDs for optical heart rate sensing with electrical skin contact measurement to ensure proper sensor contact and reduce motion artifacts [31]. The following participant data were entered: age, body mass, height, physical activity level (occasional, 0–1 h per week; regular, 1–3 h per week; frequent, 3–5 h per week; heavy, 5–8 h per week; semi-pro, 8–12 h per week; pro, 12+ h per week;—assessed via self-report), and sex. According to the manufacturer, the algorithm uses these inputs—along with measured resting heart rate and heart rate variability—to estimate $\dot{V}$O_2_max [21]. After synchronizing the sensor with Polar Flow©, it was randomly assigned to either the dominant or the non-dominant wrist, in accordance with the manufacturer’s instructions. Randomization was performed by coin toss to minimize potential bias, as the dominant wrist may differ from the non-dominant one in terms of muscularity. Participants were instructed to lie down in a supine position on the examination table, remaining still. The PFT was initiated by the study personnel, and the participant remained in supine position until the test was completed after approximately 5 min. The Polar Vantage M2, which demonstrated valid heart rate measurement via photoplethysmography (PPG) with a mean absolute percentage error (MAPE) of 4.7% [95% CI: 4.7, 4.7] in our previous validation study [27], subsequently displayed a $\dot{V}$O_2_max value that was recorded for analysis.

#### 2.3.3. Treadmill Test CPET

Following the PFT, a 12-lead ECG (Amedtec ECGpro, Aue, Germany) was attached to the participants’ upper body and participants undertook an incremental treadmill test (Venus 200/75, h/*p*/cosmos sports and medical GmbH, Nussdorf-Traunstein, Germany) using the modified Bruce protocol with stepwise increase in speed and incline [32]. Participants were allowed to choose between walking or running and were encouraged to exercise until maximal exhaustion. Ventilation and gas exchange were continuously measured throughout the test using a breath-by-breath metabolic analyzer (Metalyzer 3B, Cortex Biophysik GmbH, Leipzig, Germany). Immediately after the test, rate of perceived exertion (RPE) was assessed using the BORG scale [33]. The exercise test was followed by a 5 min cool-down phase at a walking speed of 3 km/h. $\dot{V}$O_2_max was then determined based on the highest moving 30 s averages of $\dot{V}$O_2_.

#### 2.3.4. Secondary Criteria Used to Validate $\dot{V}$O_2_max

$\dot{V}$O_2_ was considered maximal (i.e., $\dot{V}$O_2_max) if a leveling off in $\dot{V}$O_2_ was observed or if at least two of the following four criteria were met: blood lactate concentration ≥ 7 mmol/L, maximal heart rate within 10 beats of the predicted maximum heart rate (208 − 0.7 × age), respiratory exchange ratio (RER) ≥ 1.05, or a Borg scale rating ≥ 17 a.u. [14,34,35,36,37]. For justification of these criteria, we refer the reader to the discussion.

#### 2.3.5. 6-Minute Walk Test

On the following day, the 6MWT was conducted outdoors on the premises of University Hospital Salzburg according to current guidelines [24]. A flat, 50 m lap was measured and designated for the test. Upon receiving the start signal, participants were instructed to walk back and forth along the track as quickly as possible for a duration of 6 min. The total distance covered was calculated by counting the number of completed laps multiplied by the lap distance, with any partial lap distance added that was measured using a hodometer (Bosch Professional GWM32 Messrad, Bosch Power Tools GmbH, Stuttgart, Germany). After the test, participants’ RPE was assessed using Borg scale.

#### 2.3.6. 6-Minute Walk Test Based $\dot{V}$O_2_max Estimation Equations

We identified five 6MWT–based estimation equations from the literature. Eligible studies were required to report 6MWT distance alongside measured $\dot{V}$O_2_max or $\dot{V}$O_2_peak values and to provide a regression equation derived from the 6MWT. In addition, the study populations needed to be comparable to ours in terms of age and pre-existing medical conditions. Detailed information is provided in Table 1.

The equations differ based on the variables included. Ross et al. [38] and Sperandio et al. [39] rely exclusively on distance, while Porcari et al. [40] equation incorporates RPE obtained during the 6MWT. In contrast, the Sagat et al. [41] and the Burr et al. [42] equations do include anthropometric data, with Burr even including resting heart rate. In terms of average age and $\dot{V}$O_2_max, the participants from the studies by Porcari and Sagat most closely resemble ours.

**Table 1 sensors-25-05649-t001:** Overview of studies that convert covered distance from a 6-min walk test (6MWT) to peak oxygen consumption ($\dot{V}$O_2_peak) or maximal oxygen consumption ($\dot{V}$O_2_max) using an equation. f, female; m, male; resting HR, heart rate at rest; RPE, ratings of perceived exertion; $\dot{V}$O_2_max, maximal oxygen consumption; $\dot{V}$O_2_peak, peak oxygen consumption.

Author (Year of Publication)	$\dot{V}$O_2_max Estimation Equation
**Burr et al. [42]** **(2011)**	$\dot{V}$O_2_max = 70.161 + (0.023 × 6MWT [m]) − (0.276 × weight [kg]) − (6.79 × sex [m = 0, f = 1]) − (0.193 × resting HR [bpm]) − (0.191 × age [y])
**Porcari et al. [40]** **(2021)**	MaxMETs = 0.882 + (0.018 × 6MWT [m]) − (0.308 × RPE); $\dot{V}$O_2_max calculated by multiplying METs by 3.5
**Sagat et al. [41]** **(2023)**	$\dot{V}$O_2_max = 59.44 − 3.83 × sex [1 = men; 2 = women]) − 0.56 × age [years] − 0.48 × BMI [kg/m^2^] + 0.04 × 6MWT [m]
**Ross et al. [38]** **(2010)**	$\dot{V}$O_2_peak = 4.948 + 0.023 × 6MWT [m]
**Sperandio et al. [39]** **(2015)**	$\dot{V}$O_2_peak = −2.863 + (0.0563 × 6MWT [m])

### 2.4. Statistics

Descriptive statistics are given as arithmetic mean ± standard deviation (SD) or as absolute numbers and percentages if frequencies are reported. Various statistical measures were computed to assess the validity of the estimates relative to the CPET.

Statistical measures for validity of PFT and 6MWT estimates included mean absolute error (MAE, expressed as mean absolute difference in all values between CPET and $\dot{V}$O_2_max estimates) [43], MAPE (expressed as the mean absolute percentage error between CPET and $\dot{V}$O_2_max estimates across all recorded values) [44,45], Pearson correlation coefficient (r), coefficient of variation (CV, defined as the ratio of standard deviation divided by the mean) [46] and the two–way mixed model, absolute agreement intraclass correlation coefficient (ICC) [29]. Bland–Altman plots were employed to provide a graphical representation of the agreement and potential systematic biases between CPET and $\dot{V}$O_2_max estimates [44]. Validity for estimates was considered high if the MAPE was within 10% [47,48] or MAE within 3.5 mL/min/kg [19]. For justification, we refer the reader to the Discussion Section. The classification of validity based on ICC involved four categories: poor < 0.50, moderate 0.50–0.75, good 0.75–0.90, or excellent > 0.90 [29]. Statistical significance was set at *p* < 0.05. All analyses were conducted using IBM SPSS (Version 29.0.1.0, Armonk, NY, USA, IBM Corp).

Finally, to assess whether sex or the use of heart rate-limiting medication contributes to the inaccuracy of the PFT explorative, hypothesis generating sub-analyses were performed, and differences between CPET and both PFT and Porcari were calculated accordingly.

## 3. Results

### 3.1. Participants

A total of 32 participants were initially enrolled in the study and completed all testing procedures. However, data of eight participants were excluded due to failure to meet at least two of the four objective maximal exhaustion criteria. Hence, data of *n* = 24 participants were finally included in the study. Their baseline characteristics are given in Table 2. $\dot{V}$O_2_max ranged between 20.4 and 52.2 mL/min/kg, and the 6MWT distance ranged from 605 m to 885 m.

All participants were either enrolled in or scheduled to begin a facility-based medically supervised exercise training. Among these there were nine healthy subjects. Of the 15 patients, 10 patients suffered from various cardiovascular conditions (coronary artery disease (*n* = 8), valvular heart disease (*n* = 1), and congenital heart disease (*n* = 1). Furthermore, 1 participant suffered from a metabolic disease, 1 had an oncologic disease, and 3 suffered from a post–COVID-19 condition. Seven patients were taking HR limiting medication (six taking beta blockers, one taking ivabradine). The median score of the skin color tone on the Fitzpatrick scale was 3 (15 participants). A total of 79% (19 participants) had lighter skin tones (Fitzpatrick 1–3), while 21% (5 participants) had darker skin tones (Fitzpatrick 4–6).

### 3.2. Validity of $\dot{V}$O_2_max Estimation by PFT and 6MWT-Based Equations

Figure 1 presents the distribution of $\dot{V}$O_2_max for CPET, PFT, and the five estimation equations based on 6MWT. For PFT and Porcari, the median was close to CPET with −1.0 mL/min/kg and 0.2 mL/min/kg, respectively.

Table 3 summarizes the comparison of $\dot{V}$O_2_max values obtained from CPET, PFT, and 6MWT. The MAE of $\dot{V}$O_2_max estimates based on 6MWT vs. CPET ranged 4.0 mL/min/kg (Porcari) to 12.3 mL/min/kg (Ross). The lowest MAPE was observed for the Porcari equation (12.6%), while the highest MAPE was calculated for the Ross equation (34.0%). Only the Porcari equation demonstrated both a smaller MAE and MAPE, and a similar ICC compared to the PFT. The mean bias, visually indicating a systematic error, ranged from −6.4 to 12.3 mL/min/kg. See Table 3 for more details.

Figure 2 shows the Bland–Altman plots illustrating the absolute difference in PFT (Figure 2A) and the 6MWT derived estimation equation results (Figure 2B–F) compared to CPET. The mean bias for both PFT and Porcari are close to zero, while the other estimation equations clearly either systematically over- or underestimated $\dot{V}$O_2_max. The width of the limits of agreement was similar for PFT and all estimation equations, ranging from 16.9 to 26.2 mL/min/kg. Among the equations, significantly increasing overestimation of $\dot{V}$O_2_max with increasing $\dot{V}$O_2_max were observed for Porcari, Burr, Ross, and Sperandio (Figure 2). No such trend was observed for PFT and Sagat in the Bland—Altman plots. Numerical data for mean bias and limits of agreement are given in Table 3.

### 3.3. Exploratory Analysis of Determinants Contributing to the Inaccuracy of the PFT: Sex

Figure 3 visualizes the median and distribution of $\dot{V}$O_2_max data obtained via CPET, PFT and the Porcari equation, stratified by sex. We chose to present only the Porcari equation here, because the MAPE was lowest and, consequently, the overall performance superior to the other estimates.

In male participants, the difference between $\dot{V}$O_2_max obtained from CPET and PFT was calculated as a MAE of 4.6 mL/min/kg, a MAPE of 14.4%, and an ICC of 0.746. In female participants, this difference resulted in a MAE of 3.9 mL/min/kg, a MAPE of 12.6%, and an ICC of 0.746. When comparing $\dot{V}$O_2_max derived from CPET and the Porcari equation, male participants exhibited an MAE of 4.2 mL/min/kg, a MAPE of 13.5%, and an ICC of 0.759, whereas in female participants, the MAE was 3.8 mL/min/kg, the MAPE 11.2%, and the ICC 0.639.

### 3.4. Exploratory Analysis of Determinants Contributing to the Inaccuracy of the PFT: Heart Rate-Limiting Medication

Figure 4 visualizes the comparison of $\dot{V}$O_2_max from CPET with PFT and the Porcari equation, stratified by heart rate-limiting medication status (7 participants taking heart rate-limiting medication and 17 not). Among participants taking such medication, the difference between CPET and PFT resulted in a MAE of 3.7 mL/min/kg, a MAPE of 15.6%, and an ICC of 0.592, while for those not taking medication, the difference yielded a MAE of 4.6 mL/min/kg, a MAPE of 13.0%, and an ICC of 0.715. Comparing CPET and the Porcari equation, the MAE was 4.3 mL/min/kg, the MAPE 16.9%, and the ICC 0.297 for participants taking medication, whereas for those not taking medication, the MAE was 3.9 mL/min/kg, the MAPE 10.9%, and the ICC 0.707.

## 4. Discussion

The primary objective of this study was to evaluate the validity of $\dot{V}$O_2_max estimation provided by the PFT compared to the criterion measure CPET and five 6MWT-based $\dot{V}$O_2_max estimates. In line with current recommendations [19] and contrasting with existing validation studies, this study included older participants attending or intending to attend a medically supervised exercise training program at an outpatient center, including participants taking heart rate-limiting medication. This is particularly relevant, as such age and medications directly influence heart rate and potentially the accuracy of the PFT. Thus, this study holds promise to inform decision-making in preventive and rehabilitation medicine.

Our results indicate that both the PFT (MAPE 13.7%, MAE 4.4 mL/min/kg) and the 6MWT-based Porcari equation (MAPE 12.6%, MAE 4.0 mL/min/kg) exhibit a similar and acceptable group-based error (mean bias −1.0 and 0.2 mL/min/kg, respectively), thereby suggesting viable utilization, e.g., for group comparisons in studies.

On the individual level, where it comes to personalized decision-making; however, the wide limits of agreement (±11.4 and ±9.7 mL/min/kg, respectively) as well as the MAPE and MAE exceed the pre-defined cut-off limits of 10% and 3.5 mL/min/kg, respectively. The justification for these limits that provide the basis for the clinical interpretation, is as follows: The INTERLIVE Consortium has provided expert guidelines for validating wearable devices and their $\dot{V}$O_2_max estimations proposing validity thresholds for MAE ranging 1.75–3.5 mL/min/kg [19]. In our study cohort, characterized by a mean $\dot{V}$O_2_max of 33.7 mL/min/kg, a MAPE of 10% corresponds well to the upper limit proposed by INTERLIVE of 3.5 mL/min/kg. Clinically, a cut-off MAPE of 10%—corresponding to a MAE of 3.5 mL/min/kg for our study cohort—would allow for detecting changes at least close to the minimum clinically important difference, as such an improvement of 1 MET or 3.5 mL/min/kg that has been associated with a 14% reduction in all-cause mortality [6] and a 10–25% increase in survival [10]. This 3.5 mL/min/kg-level of accuracy would also suffice for cross-sectional assessments such as cardiorespiratory fitness evaluation and prognostic classification into clinically meaningful categories of 4–6 mL/min/kg as suggested by Arena et al. [49]. Furthermore, longitudinal changes such as intervention-related improvements after 15 months (4.3 mL/min/kg) [50] or age-related declines over 15 years (–5.2 mL/min/kg) [51] would be clearly detectable. However, even these error margins of 10% MAPE and 3.5 mL/min/kg MAE are too large to capture typical short-term training-induced improvements in $\dot{V}$O_2_max (2.1–3.1 mL/min/kg) over 4 weeks [52] and are entirely inappropriate for patients awaiting heart transplantation ($\dot{V}$O_2_max ≈ 10–14 mL/min/kg) [53], where substantially higher accuracy is required. Therefore, as emphasized by INTERLIVE [19], healthcare professionals must ultimately define appropriate cut-off thresholds depending on the intended application, such as monitoring training zones in athletes versus use in patients with cardiovascular disease. Of note, the error in the present study exceeded both the 10% MAPE and the 3.5 mL/min/kg MAE thresholds, underscoring the limited accuracy of the PFT.

Of note, the INTERLIVE meta-analysis reported poorer results for PFT testing, with a mean bias of 2.2 mL/min/kg and wider limit of agreements of ±15.2 mL/min/kg [19] compared to the present study. Polar^®^ claims that the mean prediction error of the PFT for $\dot{V}$O_2_max ranges between 8% and 15% [21]. However, the Polar white paper does not provide detailed insights into the statistical calculation method of the mean error, necessitating cautious interpretation of these results. Additionally, a recent study by Hansen et al. [54] reported a higher MAPE and mean bias of 14.8% and 4.4 mL/min/kg of the PFT in a population of 20 healthy participants with a mean age of 47 ± 7 years. The INTERLIVE Consortium meta-analysis demonstrated that exercise-based wearable $\dot{V}$O_2_max estimation shows lower systematic and random errors (mean bias −0.1 mL/min/kg and limits of agreement ± 9.8 mL/min/kg) compared to the PFT (mean bias 2.2 mL/min/kg and limits of agreement ± 15.2 mL/min/kg) [19]. In the present study, the PFT achieved similar performance (mean bias −1.0 mL/min/kg and limits of agreement ± 11.4 mL/min/kg), closely aligning with the accuracy reported for exercise-based tests in the meta-analysis. Future research should examine whether incorporating exercise-based wearable $\dot{V}$O_2_max estimations enhances measurement accuracy in this cohort. None of the studies mentioned used the Polar Vantage M2 PPG-based smartwatch for the PFT. This is important to acknowledge, because the PPG sensor quality may vary between devices and therefore influence the accuracy of heart rate and heart rate variability measurement, on which PFT relies on.

The necessity of applying objective criteria to verify whether participants have reached maximal exertion during CPET is a critical consideration in $\dot{V}$O_2_max validation studies [19]. This step is essential to ensure the accuracy of $\dot{V}$O_2_max measurements of the reference method, particularly when evaluating estimation tools such as the PFT. Given the characteristics of our study population—older adults, including individuals taking heart rate-limiting medication—we carefully applied established criteria to confirm $\dot{V}$O_2_max. A $\dot{V}$O_2_ leveling-off, defined as a failure to increase $\dot{V}$O_2_ by at least 150 mL/min despite a further rise in workload [2], was considered the primary indicator. Otherwise, maximal exertion was assumed if at least two of the following four criteria were met: blood lactate concentration ≥ 7 mmol/L, heart rate within 10 beats of the age-predicted maximum (208 − 0.7 × age), RER ≥ 1.05, or a Borg RPE score ≥ 17 a.u. [14,34,35,36,37]. Applying these criteria led to the exclusion of eight participants, highlighting the methodological rigor and caution taken to ensure that only valid $\dot{V}$O_2_max data were used for evaluating the PFT’s accuracy.

### 4.1. 6MWT Equation-Based $\dot{V}$O_2_max Estimates

The second aim of the study was to compare CPET and PFT with $\dot{V}$O_2_max estimation equations based on the clinically accepted 6MWT, either including solely distance (Ross, Sperandio) [38,39] or additional measures (Burr, Porcari, Sagat) [40,41,42].

Our findings indicate that the Porcari equation demonstrated a lower absolute mean bias (0.2 mL/min/kg) compared to the PFT (−1.0 mL/min/kg). Considering that the Porcari equation incorporates exercise performance data from the 6-min walk test (6MWT), whereas the PFT estimates $\dot{V}$O_2_max at rest, this result is consistent with findings from the INTERLIVE Consortium. The consortium reported that exercise-based estimations of $\dot{V}$O_2_max provide more accurate results than those derived from resting tests, at least in healthy, young populations [19]. The remaining 6MWT-based $\dot{V}$O_2_max estimation equations demonstrated lower validity and were also inferior in terms of mean error and limits of agreement compared to the PFT. The superiority of the Porcari equation may be attributed to the integration of RPE, a variable that adds a measure of intensity, which is not considered by other equations. Furthermore, the study population of Porcari and colleagues [40] is most similar in terms of age and patients’ clinical symptoms to our study. The Ross equation, in contrast, was calculated in a very different population of heart or lung transplant patients. The average $\dot{V}$O_2_max in these patients ranged from 9.6 to 16.8 mL/min/kg, which is considerably lower compared to our participants. Noteworthy, the Ross equation had the least accurate results. This underlines that researchers need to take into account the specifics of the target population when selecting estimation equations [38].

### 4.2. Exploratory Analysis by Sex and Heart Rate–Limiting Medication

Although the current sample is relatively small and exhibits an uneven distribution, we deemed it worthwhile to conduct an exploratory sub analysis by sex and heart rate-limiting medication. This decision is based on the fact that neither the cohort used to develop the PFT algorithm, nor the algorithm itself has been publicly disclosed. Consequently, it remains unclear whether the estimation of $\dot{V}$O_2_max is equally valid across sexes and use of heart rate-limiting medication. While the present analyses is purely hypothesis-generating and not powered for definitive subgroup comparisons, it may nonetheless provide useful initial insights that could inform future, more targeted research.

For both, PFT and Porcari, $\dot{V}$O_2_max estimates for female participants were closer to CPET than for male participants indicating higher validity for females. PFT and Porcari estimates yielded similar validity; however, even for female participants the results exceeded the a priori defined validity cut-off threshold of 10% MAPE or 3.5 mL/min/kg MAE.

### 4.3. Study Limitations

As discussed earlier, we are aware of the very small and unevenly distributed sample size regarding sex and medication. Therefore, we consider the conducted sub analysis stratified by sex and medication to be hypothesis-generating and exploratory. The findings should be interpreted with caution and require further investigation in larger, more balanced studies to draw definitive conclusions. Additionally, another limitation of this study is that cross-sectional data were collected based on a single measurement per participant. Because repeated measurements were not performed, the test–retest reliability of the PFT remains unknown. Accordingly, the typical error and the sensitivity of the PFT to detect true changes in $\dot{V}$O_2_max after an intervention could not be established, which should be examined in future research. Additionally, future research should include longitudinal or home-based validation studies, investigate other clinical subgroups such as frail older adults aged 60 to 90 years.

## 5. Conclusions

With the growing use of smartwatches and wearables to monitor physical activity and assessing fitness, a valid, low-effort $\dot{V}$O_2_max estimation method would be highly valuable also in patient populations. The PFT holds promise in this regard and our findings indicate that, at the population level, it performs well, similar to the 6MWT-based Porcari equation. However, due to its wide limits of agreement and the magnitude of MAE and MAPE, the applicability of PFT for personalized decision-making is very limited. Therefore, CPET remains the standard for personalized CRF assessment.

## Figures and Tables

**Figure 1 sensors-25-05649-f001:**
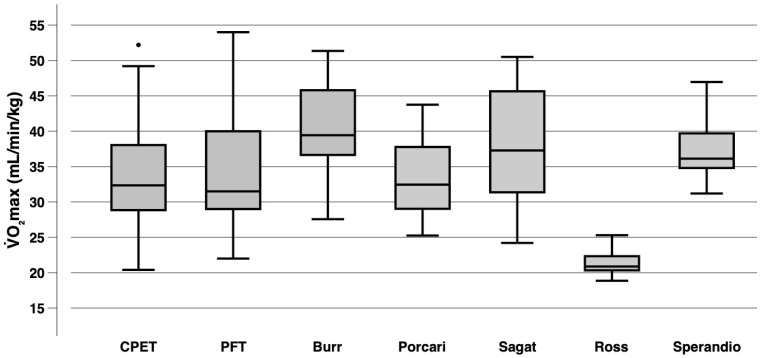
The *X*-axis represents the method of maximal oxygen consumption ($\dot{V}$O_2_max) estimation, with cardiopulmonary exercise testing (CPET) serving as the criterion measurement. Whiskers represent the minimum and maximum. Mild outliers, defined as values 1.5 to 3 times the interquartile range above the third quartile or below the first quartile, are shown as circles. The horizontal line within each box indicates the median, while the first and third quartiles define the interquartile range. Outliers are illustrated as individual dots outside the whiskers. Burr et al. [42] (2011); PFT, Polar Fitness Test; Porcari et al. [40] (2021); Ross et al. [38] (2010); Sagat et al. [41] (2023); Sperandio et al. [39] (2015).

**Figure 2 sensors-25-05649-f002:**
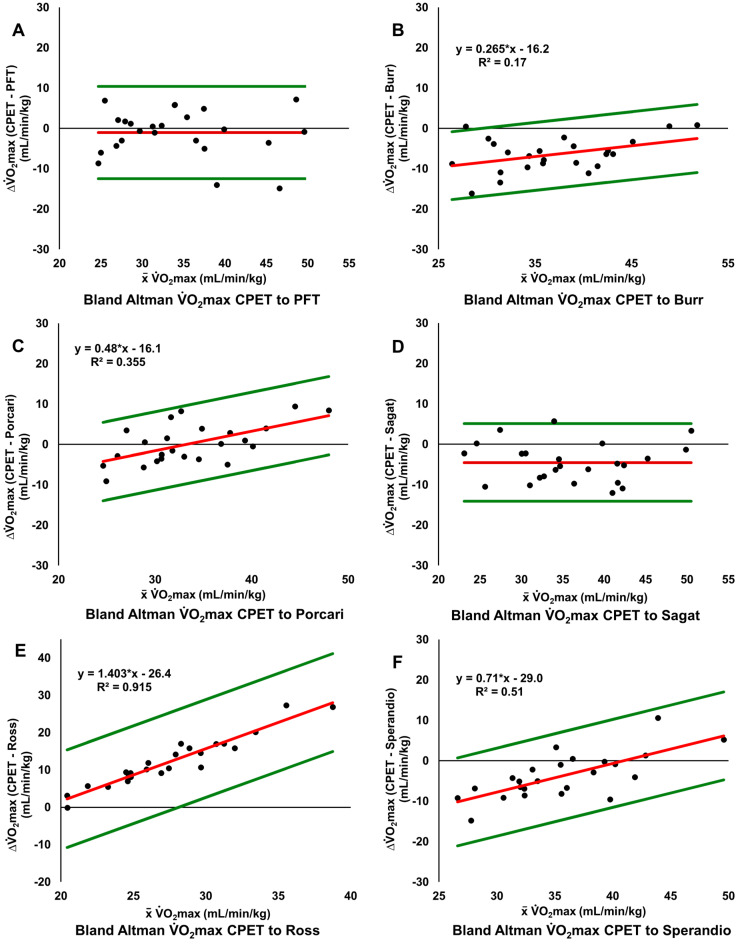
Bland—Altman plots of CPET vs. PFT and 6MWT estimation equations. The red line indicates the systematic error (mean bias), green lines indicate 95% limits of agreement calculated as means ± 1.96 × SD. Burr et al. [42] (2011); CPET, cardiopulmonary exercise test; PFT, Polar Fitness Test; Porcari et al. [40] (2021); Ross et al. [38] (2010); Sagat et al. [41] (2023); Sperandio et al. [39] (2015).

**Figure 3 sensors-25-05649-f003:**
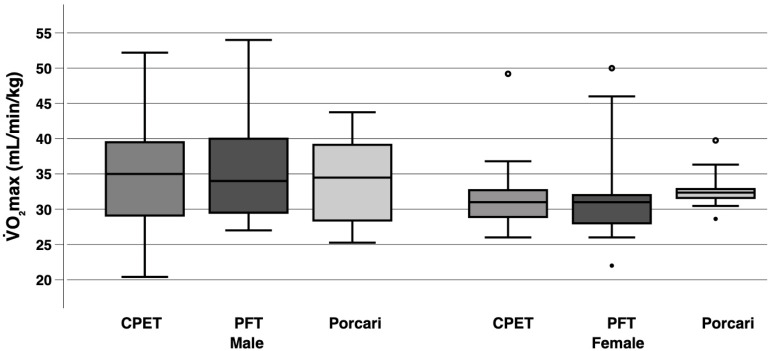
Exploratory analysis comparing $\dot{V}$O_2_max values from PFT and the Porcari equation with CPET, stratified by sex (15 male and 9 female participants). The boxplots display the minimum and maximum values (excluding outliers) through whiskers, with the median represented by a horizontal line and the interquartile range defined by the first and third quartiles. Mild outliers, defined as values 1.5 to 3 times the interquartile range above the third quartile or below the first quartile, are shown as dots. Extreme outliers, defined as values more than 3-fold the interquartile range above the third quartile or below the first quartile are marked as circles. CPET, cardiopulmonary exercise test; PFT, Polar Fitness Test; $\dot{V}$O_2_max, maximal oxygen consumption.

**Figure 4 sensors-25-05649-f004:**
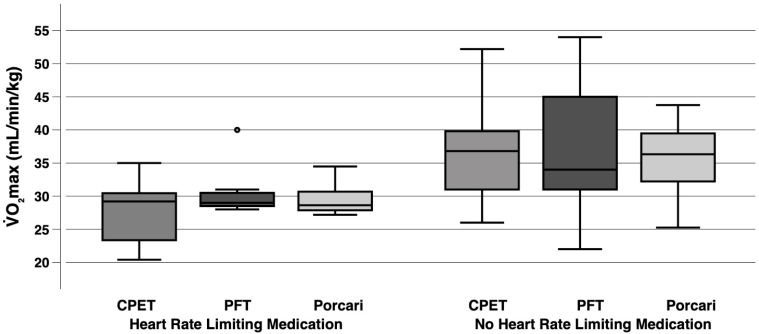
Exploratory analysis comparing $\dot{V}$O_2_max from PFT and the Porcari equation with CPET, stratified by medication status (7 participants taking heart rate-limiting medication and 17 not). The boxplots display the minimum and maximum values (excluding outliers) through whiskers, with the median represented by a horizontal line and the interquartile range defined by the first and third quartiles. Mild outliers, defined as values 1.5- to 3-fold the interquartile range above the third quartile or below the first quartile, are shown as dots. Extreme outliers, defined as values more than 3-fold the interquartile.

**Table 2 sensors-25-05649-t002:** Baseline characteristics of study participants. 6MWT distance, distance covered during 6–minute walk test; BMI, body mass index; HRmax, maximum heart rate; $\dot{V}$O_2_maxCPET, maximal oxygen consumption as measured during cardiopulmonary exercise testing. Data are given as arithmetic mean ± standard deviation. Heart rate-limiting medication, number (*n*) of participants who were on a regular regimen of β-blocker medication.

	Male (*n* = 15)			Female (*n* = 9)			Total (*n* = 24)		
**Age** (**years**)	59.5	±	9.8	54	±	10.4	57.4	±	10.2
**Height (cm)**	176.7	±	4.4	165.7	±	5.6	172.6	±	7.2
**Body mass (kg)**	84.2	±	14.4	67.3	±	13.3	77.8	±	16.0
**BMI (kg/m^2^)**	27.0	±	4.4	24.3	±	3.6	26.0	±	4.2
**$\dot{V}$O_2_max_CPET_ (mL/min/kg)**	34.3	±	8.5	32.7	±	6.9	33.7	±	7.8
**HR_max_ (bpm)**	156.2	±	20.3	171.3	±	14.9	161.9	±	19.6
**6MWT distance (m)**	730.4	±	86.4	693	±	33.8	716.4	±	72.7
**HR limiting medication (*n*)**	6			1			7		

**Table 3 sensors-25-05649-t003:** Measures of validity for maximal oxygen consumption ($\dot{V}$O_2_max) with cardiopulmonary exercise testing (CPET) representing the criterion measure vs. Polar Fitness Test (PFT), and $\dot{V}$O_2_max-estimations based on the 6-min walk test. Burr et al. [42] (2011); CPET, cardiopulmonary exercise test; CV, coefficient of variation; ICC, intraclass correlation coefficient; LoA, Limits of Agreement; MAE, mean absolute difference; MAPE, mean absolute percent error; PFT, Polar Fitness Test; Porcari et al. [40] (2021); Ross et al. [38] (2010); Sagat et al. [41] (2023); Sperandio et al. [39] (2015).

Variable	CPET	PFT	Burr	Porcari	Sagat	Ross	Sperandio
**Mean $\dot{V}$O_2_max +/− SD** (mL/min/kg)	33.7 ± 7.8	34.8 ± 8.4	40.2 ± 6.1	33.5 ± 5.0	38.2 ± 7.9	21.4 ± 1.7	37.5 ± 4.1
**MAE [95% CI]** (mL/min/kg)		4.4 [2.7, 6.0]	6.6 [4.9, 8.3]	4.0 [2.9, 5.2]	5.6 [4.1, 7.1]	12.3 [9.5, 15.1]	5.5 [4.0, 7.1]
**MAPE [95% CI]** (%)		13.7 [8.3, 19.1]	22.4 [14.8, 30.0]	12.6 [8.6, 16.7]	17.8 [12.4, 23.3]	34.0 [28.8, 39.1]	18.7 [11.9, 25.6]
**CV** (%)		24.2	15.2	15	20.6	7.8	10.9
**Mean Bias** (mL/min/kg)		−1.0	−6.4	0.2	−4.5	12.3	−3.8
**LoA** (mL/min/kg)		−12.5, 10.4	−14.9, 2.0	−9.5, 9.9	−14.1, 5.1	−0.8, 25.4	−14.6, 7.1
**ICC [95% CI]**		0.743 [0.495, 0.879]	0.575 [−0.093, 0.855]	0.725 [0.458, 0.871]	0.697 [0.137, 0.887]	0.090 [−0.073, 0.339]	0.518 [0.104, 0.769]
**Pearson Correlation Coefficient [95% CI]** ***p*-value**		0.743 [0.494, 0.882] *p* < 0.001	0.837 [0.655, 0.928] *p* < 0.001	0.788 [0.564, 0.904] *p* < 0.001	0.806 [0.597, 0.913] *p* < 0.001	0.734 [0.470, 0.878] *p* < 0.001	0.734 [0.470, 0.878] *p* < 0.001

## Data Availability

The data that support the findings of this study are openly available at Zenodo.org.

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
