# Peer review of "Validation of the Polar Fitness Test for Estimation of Maximal Oxygen Consumption at Rest in Medically Supervised Exercise Training: Comparison with CPET and the 6-Minute Walk Testâ€"

_sensors, 2025, doi:10.3390/s25185649_

Round 1
Reviewer 1 Report
Comments and Suggestions for Authors
The topic of the article is very interesting, and the recommended practical solutions take into account the specific needs of the patient and the varying needs resulting from disease factors. These solutions are essential for patients who, for various reasons, cannot engage in physical activity. Although these solutions are not perfect, as the authors emphasize, they are nevertheless very important in certain circumstances.
The article's structure is correct; it is suggested that information about the Polar Vantage M2 sensor technology be added to the introduction. The methodology is transparent, and the results are presented clearly.
Were there any individuals with disabilities in the study group? (I don't see such information.) A good discussion with sound conclusions.
From a technical perspective, the duplicated chapters 4.2 and 4.3 (lines 416-443) should be removed.
After incorporating the suggested corrections, I accept the article for publication.
Author Response
Dear Editor
Thank you for giving us the opportunity to revise our manuscript. We are grateful to the reviewers for taking their time to review our manuscript. We appreciate their diligence in feedback, which helped us to improve the clarity and quality of our manuscript. Below, we respond to each comment in detail.
Replies to Reviewer #1
Comment. The topic of the article is very interesting, and the recommended practical solutions take into account the specific needs of the patient and the varying needs resulting from disease factors. These solutions are essential for patients who, for various reasons, cannot engage in physical activity. Although these solutions are not perfect, as the authors emphasize, they are nevertheless very important in certain circumstances. After incorporating the suggested corrections, I accept the article for publication.
Response: We thank the reviewer for the positive feedback.
Comment. The article's structure is correct; it is suggested that information about the Polar Vantage M2 sensor technology be added to the introduction. The methodology is transparent, and the results are presented clearly.
Response: We thank the reviewer for this valuable suggestion regarding technical specifications of the Polar Vantage M2. In the revised manuscript, we have added information about the underlying Polar Vantage M2 sensor technology. Specifically, we now state that the optical heart rate measurement relies on Polar Precision Prime™ technology, which combines multiple LEDs for optical sensing with electrical skin contact measurement to ensure proper sensor contact and reduce motion artifacts (lines 166-169).
Comment. Were there any individuals with disabilities in the study group? (I don't see such information.)
Response. We thank the reviewer for raising this important point. According to the inclusion criteria (line 130), individuals with physical limitations that would prevent them from carrying out the planned examinations and tests were not included in the study. Therefore, participants with relevant physical disabilities were not enrolled.
Comment. From a technical perspective, the duplicated chapters 4.2 and 4.3 (lines 416-443) should be removed.
Response. The duplicated section has been removed.
Finally, we would like to thank the reviewer for the helpful and very valuable comments.
Reviewer 2 Report
Comments and Suggestions for Authors
Dear authors,
Your research can be of great benefit, as it can facilitate assessments to determine Vo2 max and help tailor and personalize training sessions as much as possible. Furthermore, conducting it with the study population provides even greater value, because in many cases, these individuals lack easy access to extensive resources.
However, when reading the article, it is necessary to clearly describe the objective(s) and hypothesis(es) of the study, as the last paragraph of the introduction does not clarify this information.
Minor points include:
- Line 182: Separate the number 6 from the section number.
- Line 202: Remove the authors' names Sagat and Burr.
- Line 218: The acronym MAPE is defined, but on line 158, it has already been defined.
- Sections 4.2 and 4.3 are the same text.
In the bibliographic references section, maintain the same format:
- Some references include the full journal name, while others have the abbreviated version.
- Some references include the month of publication, while others do not.
- Reference number 14 has two dates: 1985 and 2017.
- Only reference number 8 uses the DOI as a link, while the rest do not.
Best regards.
Author Response
Dear Editor
Thank you for giving us the opportunity to revise our manuscript. We are grateful to the reviewers for taking their time to review our manuscript. We appreciate their diligence in feedback, which helped us to improve the clarity and quality of our manuscript. Below, we respond to each comment in detail.
Comment. Dear authors, your research can be of great benefit, as it can facilitate assessments to determine Vo2 max and help tailor and personalize training sessions as much as possible. Furthermore, conducting it with the study population provides even greater value, because in many cases, these individuals lack easy access to extensive resources. However, when reading the article.
Comment. It is necessary to clearly describe the objective(s) and hypothesis(es) of the study, as the last paragraph of the introduction does not clarify this information.
Response. We thank the reviewer for the careful reading and constructive feedback. Since our study is evaluative rather than explanatory, with the primary aim of reporting a range of performance metrics rather than testing a single cut-off value, we believe that a clearly stated study objective is sufficient in this context. In line with the reviewer’s suggestion, we have further clarified the study objective in the revised manuscript to enhance clarity (lines 106-107).
Comment (Line 182): Separate the number 6 from the section number.
Response. We have corrected the formatting by separating the number 6 from the section number as recommended.
Comment (Line 202): Remove the authors' names Sagat and Burr.
Response. We thank the reviewer for this observation. The equations and the corresponding authors are introduced here for the first time, and therefore their full names are provided in order to define the abbreviations. This allows us to refer to them consistently using the shortened forms “Sagat” and “Burr” in subsequent sections of the manuscript.
Comment (Line 218): The acronym MAPE is defined, but on line 158, it has already been defined.
Response. We thank the reviewer for this careful observation and have removed the redundancy in line 244 accordingly.
Comment: Sections 4.2 and 4.3 are the same text.
Response. The duplicated section has been removed.
Comment. In the bibliographic references section, maintain the same format:
- Some references include the full journal name, while others have the abbreviated version.
- Some references include the month of publication, while others do not.
- Reference number 14 has two dates: 1985 and 2017.
- Only reference number 8 uses the DOI as a link, while the rest do not.
Response. We apologize for this and the previous oversight and thank the reviewer for the careful reading and attention to detail. The references have been formatted consistently now.
Finally, we would like to thank the reviewer for the helpful and very valuable comments.
Reviewer 3 Report
Comments and Suggestions for Authors
I would like to thank the authors for their well-structured and timely contribution addressing the validity of the Polar Fitness Test (PFT) in a clinical and rehabilitative context. The topic is relevant and well-aligned with the growing interest in non-invasive and remote monitoring of cardiorespiratory fitness. The manuscript is clearly written, and the methodological approach is generally sound, particularly the use of appropriate validity metrics and comparison with multiple 6MWT-based equations.
However, I encourage the authors to address a few methodological and interpretative aspects to strengthen the manuscript.
- Line 44 – Comment:
Please add a supporting reference to the statement: "The maximal volume of oxygen consumed per minute (V̇O₂max) is an important marker of cardiovascular function and cardiorespiratory fitness (CRF)." -
Lines 64–66 – Comment:
The statement regarding the limitations of CPET—"achieving maximal voluntary exhaustion on a treadmill or cycle ergometer may not be feasible for some patients due to e.g. exercise intolerance, neurological, or orthopedic limitations"—requires a proper citation. Please include a reference that supports this claim, particularly from clinical guidelines or relevant studies in special populations. - Lines 70–71 – Comment:
It is suggested to add a citation to support the statement: "In 2023, global shipments of wearable devices, mainly fitness trackers and smartwatches, exceeded 500 million units." - Lines 75–79 – Comment:
It is suggested to add a citation to support the following statement: “It is among the few available approaches that claim to provide a valid V̇Oâ‚‚max estimation at rest, without requiring physical exertion, solely based on a heart rate measurement and few personal data, thus offering a low-risk, easily accessible alternative to CPET, suitable for both in-person prehabilitation and rehabilitation settings and telerehabilitation.” - Lines 95–99 – Comment:
It is suggested to add citations to support the ideas presented in the following statement: “The distance covered—along with other metrics, depending on the equation applied—is then used to estimate V̇Oâ‚‚max based on regression equations derived from previous studies. However, compared to the PFT, it requires considerably more time and physical effort, as well as the patient's ability and motivation to walk.” - Lines 117–133 – Comment:
The sample size and sampling method are not clearly specified in the Participants section, which is typically expected for transparency and reproducibility. Moreover, the justification for the sample size based solely on comparison with previous studies is methodologically weak. I strongly suggest replacing this rationale with a formal sample size calculation—even if the final sample is smaller than ideal. As currently presented, this justification appears somewhat convenient rather than transparent, particularly given that it aligns with the method the authors themselves used. This could be perceived as a potential conflict of interest and may represent a significant methodological limitation. - PFT reproducibility – Lines 135–160:
Although the Polar Fitness Test (PFT) procedure is well described, no repeated measurements were performed to assess test-retest reliability. Given that the study aims to validate a wearable-based estimation method, the absence of reproducibility data should be acknowledged as a limitation or addressed in future work. - Sensor wrist assignment – Lines 151–152:
The random assignment of the sensor to either the dominant or non-dominant wrist is briefly mentioned, but the method of randomization is not described. Please specify how randomization was conducted and whether wrist placement was considered as a factor potentially influencing the results. -
Physical activity level input – Lines 147–148:
The manuscript states that participants’ physical activity level was entered into the Polar Flow software, but it is unclear how this variable was assessed (e.g., validated questionnaire, self-report, categorical scale). Since this information is used by the PFT algorithm, the method of its determination should be explicitly described. - VOâ‚‚max validation criteria in CPET – Lines 175–181:
The criteria used to validate whether VOâ‚‚max was reached during CPET are appropriate, but the original sources or guidelines supporting these thresholds (e.g., lactate ≥ 7 mmol/L, RER ≥ 1.05, HR within 10 beats of predicted max) should be explicitly cited to enhance transparency and methodological rigor. -
Discussion – Comment: Clinical Relevance of Validity Thresholds
The authors mention the validity thresholds (e.g., MAPE <10%, MAE <3.5 mL/kg/min) and relate them to the concept of the minimum clinically important difference. However, the clinical applicability of these thresholds in the context of rehabilitation or prehabilitation is not sufficiently addressed. Please elaborate on how exceeding these thresholds would affect real-world clinical decisions, such as exercise prescription, risk stratification, or patient monitoring.
-
Discussion – Comment: Broader Contextualization with Comparable Methods
The discussion would benefit from a broader comparison with alternative non-exercise-based VOâ‚‚max estimation methods, including other wearable devices or algorithms that have demonstrated higher validity. This would help contextualize the findings and highlight the relative advantages or limitations of the PFT in current clinical and technological practice.
-
Discussion – Comment: Interpretation of Exploratory Analyses
The exploratory analyses by sex and heart rate-limiting medication are repeated (lines 416–438), and no physiological interpretation is provided. It is recommended to avoid redundancy and instead discuss possible mechanisms (e.g., autonomic modulation, chronotropic incompetence, sex-based differences in HRV) that may explain the observed trends in estimation accuracy.
-
Discussion – Comment: Future Research Directions
The authors should consider including more specific recommendations for future research, such as conducting longitudinal or home-based validation studies, examining other clinical subgroups (e.g., frail older adults), or evaluating consistency across different generations of Polar devices. This would add practical value and demonstrate a clear path for continued investigation.
Author Response
Dear Editor
Thank you for giving us the opportunity to revise our manuscript. We are grateful to the reviewers for taking their time to review our manuscript. We appreciate their diligence in feedback, which helped us to improve the clarity and quality of our manuscript. Below, we respond to each comment in detail.
I would like to thank the authors for their well-structured and timely contribution addressing the validity of the Polar Fitness Test (PFT) in a clinical and rehabilitative context. The topic is relevant and well-aligned with the growing interest in non-invasive and remote monitoring of cardiorespiratory fitness. The manuscript is clearly written, and the methodological approach is generally sound, particularly the use of appropriate validity metrics and comparison with multiple 6MWT-based equations. However, I encourage the authors to address a few methodological and interpretative aspects to strengthen the manuscript.
Comment (Line 44): Please add a supporting reference to the statement: "The maximal volume of oxygen consumed per minute (V̇O₂max) is an important marker of cardiovascular function and cardiorespiratory fitness (CRF)."
Response. We thank the reviewer for this helpful comment. A supporting reference has been added to substantiate the statement regarding V̇O₂max.
Comment (Lines 64–6): The statement regarding the limitations of CPET—"achieving maximal voluntary exhaustion on a treadmill or cycle ergometer may not be feasible for some patients due to e.g. exercise intolerance, neurological, or orthopedic limitations"—requires a proper citation. Please include a reference that supports this claim, particularly from clinical guidelines or relevant studies in special populations.
Response We thank the reviewer for this comment. To substantiate our statement, we have now added supporting references for special populations and added clinical guidelines, namely: [Rapin et al., 2013; Wenzel et al., 2024; Guazzi et al., 2016].
Comment (Lines 70–71): It is suggested to add a citation to support the statement: "In 2023, global shipments of wearable devices, mainly fitness trackers and smartwatches, exceeded 500 million units."
Response. We thank the reviewer for this suggestion. We have revised the text to combine the statement about global wearable device shipments with the subsequent sentence, clarifying that both are supported by the same citation, improving clarity and ensuring accurate referencing. It now reads: “In 2023, global shipments of wearable devices, mainly fitness trackers and smartwatches, exceeded 500 million units and this number is expected to grow by at least 25% by 202821”
Comment (Lines 75–79): It is suggested to add a citation to support the following statement: “It is among the few available approaches that claim to provide a valid V̇Oâ‚‚max estimation at rest, without requiring physical exertion, solely based on a heart rate measurement and few personal data, thus offering a low-risk, easily accessible alternative to CPET, suitable for both in-person prehabilitation and rehabilitation settings and telerehabilitation.”
Response. We thank the reviewer for this comment. We added a Meta-analysis and Expert Statement that supports our statement [Molina-García et al., 2022].
Comment (Lines 95–99): It is suggested to add citations to support the ideas presented in the following statement: “The distance covered—along with other metrics, depending on the equation applied—is then used to estimate V̇Oâ‚‚max based on regression equations derived from previous studies. However, compared to the PFT, it requires considerably more time and physical effort, as well as the patient's ability and motivation to walk.”
Response. The statement reflects our own comparison of the two tests. Specifically, the PFT requires approximately five minutes and can be performed under resting conditions without the patient leaving the examination room, as the test is conducted while lying down. In contrast, the 6MWT requires at least six minutes of walking, plus additional time for preparation (e.g., transfer to a suitable location with a measured 30 m track, test instructions, and return), and in most cases also a recovery period afterwards. Furthermore, patient safety measures are warranted, especially for at-risk individuals. Thus, the examiner’s workload is significantly higher during the 6MWT. We conducted a literature search but did not identify direct references that make this specific comparison; therefore, we have clarified in the manuscript that this statement is based on our own assessment (lines 100-105).
Comment (Lines 117–133): The sample size and sampling method are not clearly specified in the Participants section, which is typically expected for transparency and reproducibility. Moreover, the justification for the sample size based solely on comparison with previous studies is methodologically weak. I strongly suggest replacing this rationale with a formal sample size calculation—even if the final sample is smaller than ideal. As currently presented, this justification appears somewhat convenient rather than transparent, particularly given that it aligns with the method the authors themselves used. This could be perceived as a potential conflict of interest and may represent a significant methodological limitation.
Response. We thank the reviewer for this important comment. In the revised manuscript, we have included a formal sample size calculation using the formula proposed by Bujang et al. (2017), with the intraclass correlation coefficient (ICC) as the target parameter. An expected ICC of 0.90 was defined, as reported in previous studies on similar comparisons (Molina-García et al., 2022), while a minimum acceptable ICC of 0.75 was chosen, corresponding to “good” reliability according to Koo and Li (2016). With α set at 0.05 and power at 0.80, the required sample size was calculated to be 23 participants. Our study included 24 participants, so the sample size is sufficient and not a limitation. We would like to thank the reviewer for pointing out and clarifying this strength of our study (lines 142-152).
Comment (PFT reproducibility – Lines 135–160): Although the Polar Fitness Test (PFT) procedure is well described, no repeated measurements were performed to assess test-retest reliability. Given that the study aims to validate a wearable-based estimation method, the absence of reproducibility data should be acknowledged as a limitation or addressed in future work.
Response. We thank the reviewer for highlighting this important point. As suggested, we have now clarified in the limitations section that, due to the absence of repeated measurements, test–retest reliability of the PFT could not be assessed. We also explicitly note that this should be addressed in future studies. (line: 492-497)
Comment (Sensor wrist assignment – Lines 151–152): The random assignment of the sensor to either the dominant or non-dominant wrist is briefly mentioned, but the method of randomization is not described. Please specify how randomization was conducted and whether wrist placement was considered as a factor potentially influencing the results.
Response. Thank you for pointing this out. We have now specified the method of randomization in the manuscript. Randomization of wrist placement was performed by coin toss, This procedure was applied to minimize potential bias, as the dominant wrist may differ from the non-dominant one in muscularity. The corresponding clarification has been added to the Methods section (line: 177-179).
Comment (Physical activity level input – Lines 147–148): The manuscript states that participants’ physical activity level was entered into the Polar Flow software, but it is unclear how this variable was assessed (e.g., validated questionnaire, self-report, categorical scale). Since this information is used by the PFT algorithm, the method of its determination should be explicitly described.
Response. Physical activity level has now been specified as being assessed via self-report in the Methods section. It now reads: “The following participant data were entered: age, body mass, height, physical activity level (occasional, 0-1 hours per week; regular, 1-3 hours per week; fre-quent, 3-5 hours per week; heavy, 5-8 hours per week; semi-pro, 8-12 hours per week; pro, 12+ hours per week; – assessed via self-report), and sex.” (lines 170-173)
Comment (VOâ‚‚max validation criteria in CPET – Lines 175–181): The criteria used to validate whether VOâ‚‚max was reached during CPET are appropriate, but the original sources or guidelines supporting these thresholds (e.g., lactate ≥ 7 mmol/L, RER ≥ 1.05, HR within 10 beats of predicted max) should be explicitly cited to enhance transparency and methodological rigor.
Response. We have added the corresponding references to the Methods section to support the CPET validation criteria and making the applied thresholds more easily traceable (line 206). [Poole et al., 2017; Edvardsen et al., 2014; Hesse et al., 2018; Howley et al., 1995; Tanaka et al., 2001]
Comment (Discussion – Clinical Relevance of Validity Thresholds): The authors mention the validity thresholds (e.g., MAPE <10%, MAE <3.5 mL/kg/min) and relate them to the concept of the minimum clinically important difference. However, the clinical applicability of these thresholds in the context of rehabilitation or prehabilitation is not sufficiently addressed. Please elaborate on how exceeding these thresholds would affect real-world clinical decisions, such as exercise prescription, risk stratification, or patient monitoring.
Response. We added clinical interpretation points clarifying how the MAPE and MAE cut-offs apply in practice and noting that thresholds may need adjustment for specific populations (lines 369-399).
Comment (Discussion – Broader Contextualization with Comparable Methods): The discussion would benefit from a broader comparison with alternative non-exercise-based VOâ‚‚max estimation methods, including other wearable devices or algorithms that have demonstrated higher validity. This would help contextualize the findings and highlight the relative advantages or limitations of the PFT in current clinical and technological practice.
Response. Thank you. Currently, there are no validated resting VOâ‚‚max estimation methods comparable to the PFT. Several exercise-based wearable VOâ‚‚max estimations exist, as summarized in the INTERLIVE Consortium meta-analysis. In the revised manuscript, we have included a comparison with these exercise-based methods to better contextualize our findings (lines 407-415).
Comment (Discussion –Interpretation of Exploratory Analyses): The exploratory analyses by sex and heart rate-limiting medication are repeated (lines 416–438), and no physiological interpretation is provided. It is recommended to avoid redundancy and instead discuss possible mechanisms (e.g., autonomic modulation, chronotropic incompetence, sex-based differences in HRV) that may explain the observed trends in estimation accuracy.
Response. We thank the reviewer for this valuable suggestion. The redundant section has been removed. Given the exploratory nature of the sex- and medication-stratified analyses, the very limited sample size for these subgroups, and the absence of predefined hypotheses, we believe that discussing potential physiological mechanisms (e.g., autonomic modulation, chronotropic incompetence, sex-based differences in HRV) would be overly speculative and disproportionate in scope. We have therefore focused on the core measurement results, while noting that the observed trends may warrant investigation in future studies.
Comment (Future Research Directions): The authors should consider including more specific recommendations for future research, such as conducting longitudinal or home-based validation studies, examining other clinical subgroups (e.g., frail older adults), or evaluating consistency across different generations of Polar devices. This would add practical value and demonstrate a clear path for continued investigation.
Response. We appreciate these very good and valuable suggestions. We have incorporated them in the revised manuscript (lines 494-505), namely
- longitudinal validation studies
- home-based validation studies
- reliability studies
- other clinical subgroups
- patients of age between 60 and 90
Finally, we would like to thank the reviewer for the helpful and very valuable comments.
Round 2
Reviewer 2 Report
Comments and Suggestions for Authors
Dear authors,
Thank you for accepting the proposed changes. Furthermore, the changes made to the discussion allow for a better understanding of the discussion.
After reading this new version, I would just like to highlight one minor point: removing the text from lines 39 and 40.
Best regards.
Reviewer 3 Report
Comments and Suggestions for Authors
The manuscript now reads clearly and flows well from rationale to methods, results, and clinical interpretation. The added references and guideline-based criteria substantially strengthen methodological transparency.
I especially value the formal sample size calculation (ICC framework), the precise procedural descriptions—particularly the physical activity level categorization and wrist randomization—and the crisp presentation of validity metrics with useful clinical context.
The comparative framing against 6MWT equations and exercise-based wearable estimations helps situate the contribution of the PFT in clinical and telerehabilitation settings. Figures and tables are clear and consistent with the narrative. Overall, this is an excellent revision and a clinically relevant contribution.